# Rabies in Bats (Chiroptera, Mammalia) in Brazil: Prevalence and Potential Risk Factors Based on Twenty Years of Research in the Northwestern Region of São Paulo, Brazil

**DOI:** 10.3390/vetsci10010034

**Published:** 2023-01-03

**Authors:** Ana Beatriz Garcia, Cristiano de Carvalho, Daiene Casagrande, Mirelle Andrea de Carvalho Picinato, Wagner Andre Pedro, Márcia Marinho, Luzia Helena Queiroz

**Affiliations:** 1Post-Graduation Program on Animal Health, Veterinary Medicine College of Araçatuba, São Paulo State University (UNESP), Araçatuba 16050-680, SP, Brazil; 2Department of Production and Animal Health, Veterinary Medicine College of Araçatuba, São Paulo State University (UNESP), Araçatuba 16050-680, SP, Brazil; 3Department of Preventive Veterinary Medicine and Animal Reproduction, School of Veterinary Medicine, São Paulo State University (UNESP), Jaboticabal 14884-900, SP, Brazil

**Keywords:** rabies, bats, prevalence, risk factors, food habit, season of the year

## Abstract

**Simple Summary:**

Over the course of twenty years, 6389 bats were sent for rabies diagnosis in the northwest region of São Paulo state, Brazil, and the rabies-positive cases were analyzed according to their food habits, taxonomic classification, sex and season of the year to identify if any risk factors could be detected for rabies occurrence in these mammals. A higher number of cases was detected in the dry season, in bats belonging to the Vespertilionidae family. Frugivorous bats had a greater association with positivity for rabies, whereas the variable “sex” had no association.

**Abstract:**

The number of rabies cases in bats has increased recently in Brazil and in the state of São Paulo, representing a new epidemiological scenario for this zoonosis. This study aimed to analyze the prevalence of rabies in bats according to food habits, taxonomic classification, sex and season of the year to identify possible risk factors for rabies occurrence in bats. A retrospective analysis of 6389 records of bat samples, from different municipalities of São Paulo, submitted to rabies diagnosis and taxonomic identification was carried out at the Rabies Diagnostic and Chiroptera Laboratories of Unesp Araçatuba, São Paulo, Brazil, from 1998 to 2017. Seventy-six (1.1%) positive rabies cases were detected in bats from ten species and seven genera of three families. The number of rabies-positive cases was higher in the dry season, with a significant association. The prevalence was higher in the Vespertilionidae family (37), especially *Myotis nigricans* (19) and *Eptesicus furinalis* (14). Frugivorous bats had a greater association with positivity for rabies, whereas the variable “sex” had no association. We recommend that the surveillance and control of rabies should be undertaken primarily during the dry season, especially in the Vespertilionidae family species and other species with a frugivorous food habit.

## 1. Introduction

Rabies is an anthropozoonosis responsible for the death of approximately 60,000 people worldwide in 2015, especially children from rural Asia and Africa [1]. In Latin America and the Caribbean from 1998 to 2014, 778 human cases were recorded in twenty one countries, 38% of which were transmitted by bats [2]. In Brazil, there were 208 cases of human rabies, 47% (98) transmitted by bats between 2000 and 2022 [3,4].

Of the 333 species of bats in Latin America, rabies has been detected in 22.5% (75/333) [5]. In Brazil, from 2001 to 2013, 1854 cases of rabies were recorded in bats [6], affecting at least 45 species in three families and 25 genera [5,7,8].

The number of cases and bat species diagnosed with the rabies virus has increased according to the increase in deforestation in the country for the construction of hydropower infrastructure, expansion of pastures and crops and growth of the real estate market [9]. Bats play an important ecological role because of their eating habits. Frugivorous bats eat fruits and contribute to seed dispersal, nectarivorous bats feed on the nectar of flowers and contribute to plant pollination, and insectivorous bats eat insects, contributing to their control. However, only 20 to 30% of the species of Chiroptera in Brazil present good adaptation to the anthropogenic environment [10].

The research conducted in Brazil describing the occurrence of bat rabies rarely presents the taxonomic identification of all the specimens received for diagnosis, being restricted to the species that had a positive diagnosis [6,11,12,13,14,15,16], most likely due to the difficulty of performing this identification during routine laboratory procedures. Therefore, the conclusions are based on absolute numbers and not on the ratio between the positive results and the diversity of species received for diagnosis in the studied area. 

Our hypothesis considered that when assessing the rabies positivity index, by considering the taxonomic classification of the bats involved and other variables, it is possible to analyze their importance to the cycle of rabies transmission in a given geographic region. Therefore, our purpose was to identify bat rabies’ prevalence and potential risk factors, such as taxonomic classification, food habits, sex and season of the year, associated with the predominantly positive species in the state of São Paulo during the period from January 1998 to December 2017.

## 2. Materials and Methods

We performed a retrospective analysis of 10,326 samples of different mammalian species recorded by the Rabies Diagnostic Laboratories of the School of Veterinary Medicine (FMVA) of Unesp/Araçatuba from 1998 to 2017. These samples were mostly from urban areas in 59 municipalities in the north, northwest and midwest regions of the state of São Paulo, Brazil, obtained by means of passive surveillance, as recommended by the State Rabies Control Program of São Paulo state [17].

Rabies diagnosis was carried out using the fluorescent antibody test [18] and the mouse inoculation test [19], as recommended by the World Health Organization [20]. 

Species identification was performed in the Chiroptera Laboratory FMVA/Unesp, Araçatuba, SP, Brazil; where the specimens had identification numbers and were found fixed in 10% formalin and packed in glass bottles with 70% alcohol. The identification was based on the morphological and morphometric characteristics of the bats using established keys of determination, which use measurements of the forearm, head/body, ear and dental arch; the presence, shape or absence of the interfemoral membrane and the nasal appendix; hair color; and information on other structures that can distinguish each species [21,22,23].

The prevalence of rabies in the evaluated bats was calculated across all samples (absolute prevalence) and by family (FmP), genus (GnP), species (SpP), sex, food habit and season of the year by means of the quotient of the number of positive samples for each of the variables and the total number of samples analyzed for that variable. Prevalence by sex was calculated as the ratio between the number of positive samples by sex and the total number of examined samples in which the sex could be identified.

The seasons of the year were defined according to Pedro and Taddei [24] as dry (May to September) and rainy (October to April).

Differences in the rabies prevalence according to food habits, family, genus, species, sex and season were evaluated by estimating the odds ratio (OR) obtained by the adjustment of logistic regression models. Species, gender and family were analyzed separately, as these taxonomic levels are nested. Similarly, food habits showed a high correlation with family, so they were also analyzed separately. After adjustment, each model was submitted to post hoc Tukey tests considering a 5% significance level. These analyses were carried out using the stats and multicomp packages in the R v3.4.4 software, R Foudation, Vienna, Austria [25].

A map containing the municipalities of the state of São Paulo made it possible to construct a geographic database using the MapInfo Professional program (version 7.5 SCP), MAPINFO Corporation: New York, NY, USA [26]. As an identifier between the maps and spreadsheets, the “Municipalities” field was used for association. The spatial analysis sought to describe the patterns contained in the database considering the spatial locations of the samples of the rabies-positive bats.

## 3. Results

Among the 10,326 recorded samples, 6389 (61.9%) were bats, of which 5573 (87.2%) could be identified by species, resulting in 311 (5.6%) individuals of the hematophagous species *Desmodus rotundus* and 5262 (94.4%) of nonhematophagous species. After taxonomic identification, it was also possible to identify the sex of 95% of the samples, resulting in 2804 (52.9%) males and 2494 (47%) females. Some inadequate preservation conditions made it impossible to identify the species and/or sex of some specimens or to perform the rabies diagnosis.

The evaluated bat samples came from 59 municipalities in the state of São Paulo, most of them from the northwest region, of which 15 (25.4%) had positive samples (Figure 1). The annual average number of bats received for diagnosis was 309, with a general (absolute) prevalence of 1.2% (76/6322) and a mean of five positive rabies cases per year.

The number of samples sent for diagnosis and the number of positive samples changed according to the season, with the predominance of samples (63.5%) being sent in the rainy season (October to April) and a higher absolute number of positive cases (43/76) in the dry season (May to September). Considering the prevalence (number of positive bats/total number of examined bats) in each season, a significant association was observed between seasonality and the prevalence of rabies in bats (*p* < 0.05). In the dry season, the prevalence was 1.91% (43/2261), and in the rainy season, it was 0.82% (33/4061). Therefore, the chance of positive case detection in bats (OR) was about twice as high as that in the rainy season (Table 1).

Nonhematophagous bats sent for diagnosis were distributed across four families, 19 genera and 30 species (Table 2). The greatest diversity of species was observed in the Phyllostomidae family (40%), followed by Molossidae (33.3%), Vespertilionidae (23.3%) and Noctilionidae (3.3%).

The rabies virus was detected in 76 bats distributed across three families, seven genera and 10 species. No *Desmodus rotundus* vampire bat was positive for rabies in the studied area or period (Table 2).

The prevalence of rabies according to bat family (FmP) was highest in the Vespertilionidae, with 37 positive bats (FmP 7.4%), followed by Phyllostomidae with 18 positives (FmP 2.1%) and Molossidae with 21 positive bats (FmP 0.5%).

The number of examined samples was higher for the genus *Molossus* (64%), whereas *Molossus* and *Myotis* had the same number of positive samples (25% each). The prevalence according to genus (GnP) was more often detected in *Eptesicus* (GnP 10.2%), followed by *Myotis* (GnP 8.6%) and *Artibeus* (GnP 5.8).

The number of positive rabies cases was higher for *Myotis nigricans* (25%), followed by *Artibeus lituratus* (23.7%), *Eptesicus furinalis* and *Molossus rufus* (18.4%). The other positive species (*M. molossus, E. diminutus, Eumops glaucinus, Lasiurus blossevillii, L. ega* and *Cynomops planirostris*) represented 11 cases (14.5%). On the other hand, considering the prevalence by species (SpP), higher value was observed for *Eptesicus furinalis* (SpP = 13.3%), followed by *Myotis nigricans* (SpP 8.6%) and *Artibeus lituratus* (SpP 8%), as shown in Table 2.

Considering the close links among the taxonomic levels, the great diversity of species found, the great variability in the number of examined samples and the high number of negative samples for each species, logistic regression models were used to analyze association with rabies, only up to the family level, taking into account those species with at least one positive case. Therefore, when comparing families in a pairwise manner (Table 1), significant differences were observed between them (*p* < 0.05), and the Vespertilionidae family showed a 21-fold higher chance (OR 20.9) of the occurrence of positive cases compared to the Molossidae family and a 3.6-fold higher chance (OR 3.56) in comparison with the Phyllostomidae family. Phyllostomidae, in turn, presented a six times greater chance (OR 5.87) of positive cases than Molossidae. 

Eighteen species (60%) with an insectivorous food habit were identified, and nine (30%) were frugivores, followed by nectarivores, omnivores and haematophagous bats, with one species (3.3%) each (Table 2). Of the rabies-positive bats, 58 (76.3%) belonged to species with an insectivorous food habit, and 18 (23.7%) were frugivorous. The number of examined frugivorous bats was 421, with a rabies prevalence of 4.3% (18), and among 4717 examined insectivorous bats, the prevalence was 1.2% (58). The logistic regression models indicated a statistically significant association between this variable and rabies prevalence (*p* < 0.0001), with a four times higher chance (OR 4.2) of bats with frugivorous food habits being positive for rabies (Table 1). 

The variable sex was determined in 63 (82.9%) of the positive samples, resulting in 34 (54%) males and 29 (46%) females and a prevalence of 1.2% and 1.3%, respectively, with no significant association (*p* = 0.960) between sex and rabies positivity (OR 1.17).

## 4. Discussion

According to an epidemiological survey conducted by PAHO (Pan-American Health Organization) in 2003, the Latin American countries were classified into five different epidemiologic areas considering the frequency of canine rabies in relation to surveillance efforts by state, department or province [27,28]. São Paulo state were placed in group 1, as an area free of canine rabies for more than 10 years [28]. In this way, some changes occurred concerning the measures for rabies control, mainly the canine anti-rabies vaccination campaigns, which were discontinued and substituted for by dog vaccination in pre-established locations throughout the year, and the most important focus became wild animals, mainly bats, as other variants of the rabies virus are detected every year in these animals. Therefore, passive surveillance actions such as bat sampling for laboratory analysis and bat behavior studying became even more important in this new phase of rabies epidemiology in the state of São Paulo.

We observed that the frequency of sending bat samples for rabies diagnosis and the number of positive samples were both influenced by the season. The number of samples being recorded was higher during the rainy season, and this is likely due to the increase in the temperature observed in this period, which leads to greater bat activity, mostly because of the food supply (insects, fruits and flowers). During this period, most species are in the reproductive phase, and due to the high temperatures, people frequently have open windows for a relatively long period of time, increasing the number of bats entering dwellings [29]. Therefore, the probability of inhabitants finding these mammals and reporting them to the surveillance service increases, as already observed by Almeida et al. [10].

On the other hand, in the dry season, bats were twice as likely to be rabies-positive as in the rainy season, and this may be related to the greater contact among bats in this period due to competition for water and food [24].

The frequency of bats being sent for rabies examination during the study period was higher in the Molossidae family, genus *Molossus*. This fact has already been observed in previous studies in this same geographical region [30], as well as in the western region of São Paulo state [31], in the metropolitan region of São Paulo city [9,10], in the city of Campinas [32,33] and even in other Brazilian states, such as Parana and its main city, Curitiba [29]. The bats of this genus are opportunistic in choosing their shelters, exploring a wide variety of building structures, and can be found where the spacing between the ceiling and the roof does not exceed 50 cm, including in low buildings with asbestos cement tiles [9]. The easy adaptation of these bats to the urban habitat is also related to their insectivorous diet, which includes a wide variety of beetles, cockroaches, mosquitoes, termites, Neuroptera, Dermaptera, Orthoptera and Odonata, which are widely found in urban areas due to excess organic material and presence of luminosity, which attracts some of these insects [34].

Among the seven species of nonhematophagous bats commonly found in the metropolitan region of São Paulo [9], only three were frequently found in our study region (northwest of São Paulo state). This diversity of species found in different geographical regions is due to the intense dynamics of urban bat colonies, which vary according to the food supply, the number of individuals and the size of the shelter chosen.

In our 20-year retrospective study, rabies was most often found in the bat family Vespertilionidae (48.7%), as previously described in the same region [6,15] as well as in other geographical regions of São Paulo state [10,32]. This highest prevalence can be justified by the behavior pattern of this family, which can be the highest risk factor for the transmission of the rabies virus when compared to the other families [33]. This family also presented a higher FmP (7.4%) than the other families, with a chance of positivity that was 3 to 21 times higher than that shown by the Molossidae and Phyllostomidae families, respectively. This index has not been previously described and interpreted by other authors, as it is calculated taking into account the total number of sent samples specific to this family, which provides a more precise analysis when compared to the analysis of positives among the total sent samples.

In other regions of São Paulo state, in different periods, other families stand out with higher positivity [11,16,30]. This variation could be explained by the different types of urbanization found in the state, leading to changes in the food supply and shelters for these animals.

Although it had the largest number of sent samples (74.4%), the Molossidae family had the lowest number of positive samples, which was already observed in other studies carried out in the state of Paraná [29], in the city of São Paulo/SP [9] and in the city of Campinas [32]. The species of the family Vespertilionidae represented only 9.4% of the species from urban areas sent for diagnosis in the present work, as also described in the metropolitan region of São Paulo [9].

In Campinas, a big city in São Paulo state, Dias et al. [33] found a higher positivity for the rabies virus in bats of the Phyllostomidae family, and, similarly to our study, the only species of this family with a positive result was *Artibeus lituratus*. Some studies performed in other Brazilian states, such as Pará [35] and Rio de Janeiro [13], also described a higher positivity for rabies in bats of the Phyllostomidae family. This result could be explained by two factors: (1) Phyllostomidae exhibit reduced fitness because they are larger and heavier, requiring a more critical energy balance; (2) most species in this family are polyestric, and females are therefore more susceptible to seasonal changes in climate and food availability, so they have a dispersal behavior, with daily change of perch by females [36], which can increase contact with other individuals, including other species.

On the other hand, Ribeiro et al. [29], in the state of Paraná, found a higher rabies positivity in the family Molossidae. This difference in prevalence in Chiroptera families and Brazilian states is expected due to the great biodiversity found in countries with continental dimensions such as Brazil. Studies of bat rabies prevalence are commonly performed considering two main groups, hematophagous and nonhematophagous bats, and this classification contributed to the scarcity of information regarding the prevalence according to species [9].

Regarding genera, the rabies virus was detected in seven bat genera, and the percentage of positive specimens (25% each) was higher in the *Molossus* (Molossidae family) and *Myotis* (Vespertilionidae family) genera, confirming the diversity previously observed in the region [6,15,30]. Bats were usually collected and sent by virtue of changes in their physiological state, being generally found fallen on the ground during the day, so this limitation might have influenced the results. However, this is a bias common to all studies that investigate rabies in urban bats, since the capture of bats to obtain samples has ethical and environmental restrictions [9].

Among the ten positive species, the absolute rabies prevalence was higher for *Myotis nigricans* (25%). However, the SpP was higher for *Eptesicus furinalis* (13.3%), representing 18.4% of the total positive samples, corroborating the index calculated in previous studies by the same research group over a smaller time interval [30].

In the metropolitan region of São Paulo, Campinas and Botucatu, Almeida et al. [12], De Lucca et al. [32] and Souza et al. [37] highlighted the positivity in the species *Tadarida brasiliensis*. However, only three specimens of this species were found in our study, all negative for the rabies virus, even though it is a species that has a habit of living in colonies with more than 1000 individuals. Pacheco et al. [9], when calculating the constancy of bat species in urban areas, classified this species as being of common presence in the capital and its metropolitan regions of São Paulo, Paraná and Rio Grande do Sul.

None of the *Desmodus rotundus* vampire bats were positive for rabies in the present study, and the number of sent samples of this species was also lower when compared to the others, because their collection and shipment depends on the rabies control teams of the State Secretaries of Agriculture, which are responsible for the investigation of outbreaks in herbivores. When examining samples from regions more rural and closer to forests in the Brazilian state of Pará [36], a greater absolute prevalence in this species was observed.

Our results show that bat food habits were statistically associated with rabies prevalence (Table 1), differing from the results observed by De Lucca et al. [32]. This difference can be explained by the geographical area considered in each of the studies, because one study refers to a specific area in the city of Campinas/SP [33], while the current study reports the prevalence in the northwest state region with a larger number of specimens in different municipalities. In absolute numbers, the positivity was higher in bats with insectivorous food habits, as already shown in other studies [6,10,16,30]; however, according to food habits, the prevalence was higher for the frugivorous bat species, corresponding to 4.3% (18/421). This result was likely because insectivorous bats are easier to find in urban centers [8] than frugivorous bats, which need trees, such as mango and fig trees, for food and shelter.

The sex variable was not indicated as a risk factor for rabies positivity, as already described by Cabral et al. [14] in Rio de Janeiro and Souza et al. [37] in the region of Botucatu/SP.

## 5. Conclusions

The bat rabies prevalence in northwestern São Paulo state in the studied 20 years (1998 to 2017) was 1.2%, which is restricted to nonhematophagous bats in urban areas with frugivorous and insectivorous food habits, especially the species *Myotis nigricans* and *Eptesicus furinalis*. The prevalence was higher in the dry season. Strategies for the surveillance and control of bat rabies should be performed primarily during the dry season, with special attention to the two Vespertilionidae species most frequently affected.

## Figures and Tables

**Figure 1 vetsci-10-00034-f001:**
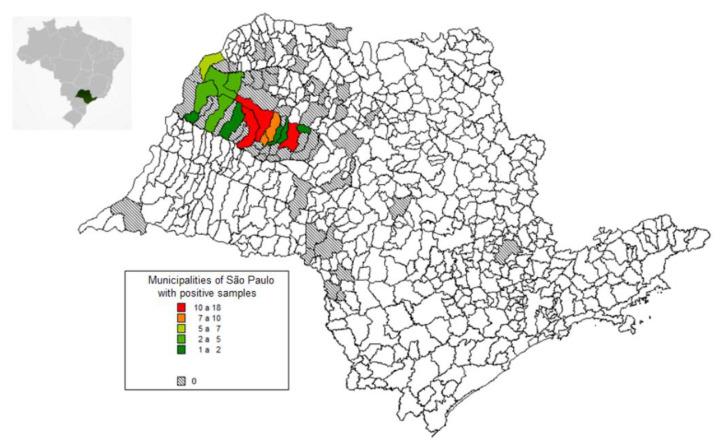
Maps of Brazil and the state of São Paulo showing the origin of the bat samples sent for rabies diagnosis from 1998 to 2017 (gray) and the cities with positive samples (colors).

**Table 1 vetsci-10-00034-t001:** Estimation of the odds ratio (OR) by logistic regression for the occurrence of positive rabies cases in bats according to food habits, family, sex and season of the year.

Variable		OR	Default Error	*p* Value
Food habit	Insectivorous Frugivorous	1 4.20	1.341	<0.0001
Family	Molossidae Phyllostomidae	1 5.87	1.442	0.001
	Molossidae Vespertilionidae	1 20.9	1.389	0.001
	Phyllostomidae Vespertilionidae	1 3.56	1.374	0.001
Season	Rainy Dry	1 2.33	1.312	0.009
Sex	Female Male	1 1.17	1.296	0.960

**Table 2 vetsci-10-00034-t002:** Number and percentage of bats sent to the Unesp Araçatuba Rabies Laboratory from 1998 to 2017 according to taxonomic classification, results and prevalence *.

Family, Gender and Species (Food Habit **)	Sample n (%)	Positive n (%)	* FmP, GnP and SpP
**Molossidae**	**4149 (74.4)**	**21 (27.6)**	**0.5**
* Cynomops *	29 (0.5)	1 (1.3)	3.4
*Cynomops planirostris* (I)	29 (0.5)	1 (1.3)	3.4
* Eumops *	524 (9.4)	1 (1.3)	0.2
*Eumops auripendulus* (I)	11 (0.2)	0	0
*Eumops glaucinus* (I)	474 (8.5)	1 (1.3)	0
*Eumops perotis* (I)	39 (0.7)	0	0.2
* Molossops *	8 (0.1)	0	0
*Molossops temminckii* (I)	8 (0.1)	0	0
* Molossus *	3565 (64,0)	19 (25)	0.5
*Molossus molossus* (I)	1620 (29.1)	5 (6.6)	0.3
*Molossus rufus* (I)	1945 (34.9)	14 (18.4)	0.7
* Nyctinomops *	20 (0.4)	0	0
*Nyctinomops laticaudatus* (I)	18 (0.3)	0	0
*Nyctinomops macrotis* (I)	2 (0.04)	0	0
* Tadarida *	3 (0.05)	0	0
*Tadarida brasiliensis* (I)	3 (0.05)	0	0
**Noctilionidae**	**43 (0.8)**	**0**	**0**
* Noctilio *	43 (0.8)	0	0
*Noctilio albiventris* (I)	43 (0.8)	0	0
**Phyllostomidae**	**856 (15.4)**	**18 (23.7)**	**2.1**
* Artibeus *	308 (5.5)	18 (23.7)	5.8
*Artibeus jamaicensis* (F)	23 (0.4)	0	0
*Artibeus lituratus* (F)	231 (4.1)	18 (23.7)	8
*Artibeus obscurus* (F)	1 (0.02)	0	0
*Artibeus planirostris* (F)	53 (0.9)	0	0
* Carollia *	50 (0,9)	0	0
*Carollia perspicillata* (F)	50 (0,9)	0	0
* Chiroderma *	5 (0.1)	0	0
*Chiroderma doriae* (F)	5 (0.1)	0	0
* Desmodus *	311 (5.6)	0	0
*Desmodus rotundus* (S)	311 (5.6)	0	0
* Glossophaga *	107 (1.9)	0	0
*Glossophaga soricina* (N)	107 (1.9)	0	0
* Phyllostomus *	17 (0.3)	0	0
*Phyllostomus hastatus* (O)	17 (0.3)	0	0
* Platyrrhinus *	44 (0.8)	0	0
*Platyrrhinus lineatus* (F)	44 (0.8)	0	0
* Uroderma *	9 (0.2)	0	0
*Uroderma bilobatum* (F)	9 (0.2)	0	0
* Sturnira *	5 (0.1)	0	0
*Sturnira lilium* (F)	5 (0.1)	0	0
**Vespertilionidae**	**525 (9.4)**	**37 (48.7)**	**7.4**
* Eptesicus *	157 (2.8)	16 (21)	10.2
*Eptesicus brasiliensis* (I)	13 (0.2)	0	0
*Eptesicus diminutus* (I)	39 (0.7)	2 (2.6)	5.1
*Eptesicus furinalis* (I)	105 (1.9)	14 (18.4)	13.3
* Lasiurus *	148 (2.7)	2 (2.6)	1.3
*Lasiurus blossevillii* (I)	51 (0.9)	1 (1.3)	2
*Lasiurus cinereus* (I)	21 (0.4)	0	0
*Lasiurus ega* (I)	76 (1.4)	1 (1.3)	1.3
* Myotis *	220 (3.9)	19 (25)	8.6
*Myotis nigricans* (I)	220 (3.9)	19 (25)	8.6
**TOTAL**	**5573**	**76**	**1.4**

* FmP = Prevalence by family; GnP = Prevalence by genus; SpP = Prevalence by species. ** I = insectivorous; F = frugivorous; S = sanguivorous; N = nectarivorous; O = omnivorous. Bold highlights the Family and underlining highlights the Genus and their corresponding total numbers.

## Data Availability

Not applicable.

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
