# Peer review of "Rabies in Bats (Chiroptera, Mammalia) in Brazil: Prevalence and Potential Risk Factors Based on Twenty Years of Research in the Northwestern Region of São Paulo, Brazil"

_vetsci, 2023, doi:10.3390/vetsci10010034_

Round 1

Reviewer 1 Report

Review of “Rabies in bats (Chiroptera, Mammalia) in Brazil: prevalence and potential risk factors”

This is a nicely done manuscript and only (very) minor revisions are needed to correct a few English word choice errors and a few places of clarification.

Line 13: English usage error: “in the last decade, from the control of rabies in dogs” should be “in the last decade, after the control of rabies in dogs”

Lines 19 and 31: In scientific writing in English, the convention is to fully write out numbers occurring at the beginning of the sentence, (I know this is a tedious point, but not following this convention really does interfere with the readability of the manuscript).

https://www.grammarly.com/blog/when-to-spell-out-numbers/#:~:text=It%20is%20generally%20best%20to,should%20be%20expressing%20numbers%20consistently.

Line 41: Over what period of time are the 60,000 deaths of people occurring?

Line 47: “Of the species of bats in Latin America, 22.5% (75/333) have been detected to be positive for rabies [5]” This sentence is a bit awkward.  Do you mean “Rabies has been detected in 75 of the 333 (22.5%) total bat species in Latin American” or “Of the 333 species of bats in Latin America, rabies has been detected in 22.5% (75/333) [5”]?

For readers less familiar with bats, I would suggest that a brief mention is made in the introduction regarding the division of bats by eating habits into hematophagous, insectivorous, and frugivorous groups.  This division is an important part of your analysis and some orientation is needed.

Line 183: “which were extinguished and substituted by fixed posts  I assume “extinguished” here means “discontinued.”  I really do not know what “fixed posts” means in the context: social media campaigns?  Occasional postmortem exams of dogs?  Something else?

Lines 206-207: “spacing with the ceiling does not exceed 50 cm” Spacing of what part of the building and the ceiling?  The ceiling and the roof (like the attic)?

I got a little lost in the comparison of the results from this study and previous ones (Lines 239-253).  Perhaps summarize and contract previous work and then compare to you own instead of going back and forth. 

The comment in Lines 251-253 (“Studies of bat rabies prevalence are commonly performed considering two main groups: hematophagous and nonhematophagous bats, and this classification contributed to the scarcity of information regarding the prevalence according to species [9].”) seems to contradict the immediately preceding comparisons of Family / Species.  Consider moving this sentence to the beginning of the section and then state something like “in previous studies that have looked at Family / species prevalence the following trends were observed” and then follow with your own results.

Line 255 Since you switch from Family to genera, consider reiterating these two bat species are in the Molossidae and Vespertilionidae Families.

Line 256 “confirming the diversity” I am not sure what is meant by “diversity” in this sentence given only two species are bats are listed.  Two species each accounting for 25% of positives each (50% total) seems like there is actually not much diversity in which bat species are affected.

Table 2:  I found this table a bit confusing at first since you break out the bats by Genera, and then underneath Genera plus species.  Consider underlying the combined Genera to better set off this difference (or some other method with consultation with the editor).

Author Response

Thanks you for your kind attention in reviewing our manuscript and for you valuable suggestions. We accepted all your suggestions and explained those one which demanded some explanation. Please find attached our responses. 

Reviewer 2 Report

This study is valuable in that it investigated and characterized the epidemiology of bat rabies in one region of Brazil over a period of 20 years.

Major and minor comments are listed below.

major comments

1. The title should be supplemented with the statement 'twenty years of research in the northwestern region of São Paulo'.

2. A simple summary should be about half.

3. Regarding the higher prevalence in the dry season than in the rainy season, I felt some discrepancy between the description of the discussion (Line 188-196) and the results of this study. If this concern of mine stems from my inability to convey what you want to say, please explain carefully. Otherwise, I think this discrepancy requires a reasonable explanation.

I think it is necessary to rule out the possibility that a particular sampling bias occurred during this passive surveillance period. (For example, large numbers of rabies-positive bats may have been introduced following a rabies outbreak in the study area during one dry season, thereby skewing the data).

In relation to this, it may be necessary to recheck whether the high prevalence in the dry season is a common trend throughout the 2 decade survey period.

If you have taken steps in your surveillance and data handling to eliminate the specific biases described above, please indicate so in the M&M section.

minor comments

1. Line 45

47.’% (98) → Correct to appropriate description

2. Line 20, 32, 298

"The highest number" → The highest-level expression is used when the number of people or things to be compared is "three or more," so modify it to an appropriate expression.

3. Correct the genus and scientific names of bats to italics

Author Response

Thanks you for your kind attention in reviewing our manuscript and for your valuable suggestions. We accepted all your suggestions, rewrote many paragraphs and explained those one, which demanded some explanation.

Round 2

Reviewer 2 Report

Now I conclude that the revised manuscript has adequately addressed all of the reviewers' comments and that the manuscript is ready for publication.